# A Novel Rexinoid Agonist, UAB116, Decreases Metastatic Phenotype in Hepatoblastoma by Inhibiting the Wnt/β-Catenin Pathway via Upregulation of *TRIM29*

**DOI:** 10.3390/ijms26093933

**Published:** 2025-04-22

**Authors:** Swatika Butey, Morgan L. Brown, Janet R. Julson, Raoud Marayati, Venkatram R. Atigadda, Maryam G. Shaikh, Nazia Nazam, Colin H. Quinn, Sorina Shirley, Laura L. Stafman, Elizabeth A. Beierle

**Affiliations:** 1Division of Pediatric Surgery, Department of Surgery, University of Alabama at Birmingham, Birmingham, AL 35233, USAmbro41@lsuhsc.edu (M.L.B.); jjulson@uabmc.edu (J.R.J.); mgshaikh@uabmc.edu (M.G.S.); nnazam@uab.edu (N.N.); chquinn@uab.edu (C.H.Q.); sorinashirley@gmail.com (S.S.); laura.stafman@childrensal.org (L.L.S.); 2Department of Dermatology, University of Alabama at Birmingham, Birmingham, AL 35233, USA; venkatra@uab.edu

**Keywords:** hepatoblastoma, retinoid X receptor (RXR), liver X receptor (LXR), Wnt/β-catenin, tripartite motif containing 29 (*TRIM 29*)

## Abstract

Hepatoblastoma (HB) is the most common pediatric primary liver tumor. About 20% of affected children have pulmonary metastasis at presentation. Survival rates for these children are dismal, not exceeding 25%. To study this subset of patients, we sequenced a metastatic HB cell line, HLM_2, and identified downregulation of the Liver X Receptor (LXR)/Retinoid X Receptor (RXR) pathway. LXR/RXRs function as transcriptional regulators that influence genes implicated in HB development, including the Wnt/β-catenin signaling pathway. We assessed the effects of a novel LXR/RXR agonist, UAB116, on metastatic HB, hypothesizing that this compound would affect genes governing the Wnt/β-catenin pathway, decreasing the metastatic phenotype of HLM_2 metastatic HB cells. We evaluated its effects on viability, proliferation, stemness, clonogenicity, and motility, and performed RNA sequencing to study differential gene regulation. Treatment with UAB116 for 72 h decreased HLM_2 proliferation, stemness, clonogenicity, and invasion. RNA sequencing identified an eight-fold increase in *TRIM29*, a gene known to inhibit β-catenin, in cells treated with UAB116. Administration of the LXR/RXR agonist, UAB116, reduces proliferation, stemness, and invasiveness of metastatic HB cells, potentially by upregulation of *TRIM29*, a known modulator of the Wnt/β-catenin pathway, providing support for further exploration of LXR/RXR agonism as a therapeutic strategy for metastatic HB.

## 1. Introduction

Hepatoblastoma (HB) is the most common pediatric primary liver tumor, typically presenting in children from 6 months to 3 years of age. HB is uncommon, occurring in approximately 1.7 cases per million children; however, its incidence has risen markedly over the past sixty years [1]. Over one-fifth of children with HB will have pulmonary metastasis present at diagnosis, which portends a particularly poor prognosis, with these children having a five-year event-free survival that does not surpass 25% [2]. There is a paucity of research on metastatic HB, primarily due to the lack of models [3]. We previously developed a metastatic hepatoblastoma cell line, HLM_2, to facilitate the study of metastatic disease [4]. Analysis of RNA sequencing data demonstrated downregulation in the Liver X Receptor/Retinoid X Receptor (LXR/RXR) pathway in these metastatic cells compared to the parent cell line. LXR/RXRs are nuclear receptors that act as transcriptional regulators of genes involved in HB carcinogenesis, including genes associated with differentiation, proliferation, cell cycle progression, and stem-cell self-renewal [5]. Additionally, LXR/RXRs are known to target genes that affect the Wnt/β-catenin signaling pathway [6], which plays a crucial role in liver growth, regeneration, and tumor development. HB has been linked to abnormal canonical Wnt/β-catenin signaling, primarily due to somatic mutations in the *CATENIN β1* (*CTNNB1*) gene [7] that are associated with an aggressive HB phenotype [8]. We recently developed a novel retinoid agonist, UAB116, that activates LXR/RXR signaling [9]. Considering the sequencing findings demonstrating downregulation of the LXR/RXR pathway in a metastatic HB cell line, we aimed to characterize the effects of this retinoid agonist on metastatic HB cells.

## 2. Results

### 2.1. LXR/RXR Pathway Is Downregulated in HLM_2 Cells

The HLM_2 cells have been extensively characterized [4]. We performed RNA sequencing on the HLM_2 cells and their parent cell line, HuH6, and analyzed the sequencing results employing Ingenuity Pathway Analysis (IPA) using the parameters of differentially expressed genes with an adjusted *p*-value below 0.05 and fold change cut-off of greater or less than two. We found pathways involved with chemokine signaling and embryonic stem cell pluripotency to be significantly upregulated in the HLM_2 metastatic HB cells compared to the parent HuH6 cells (Figure 1a, orange bars). In contrast, pathways including LXR/RXR activation, apoptosis, and senescence were significantly downregulated (Figure 1a, blue bars). RXRs are important nuclear receptors that modulate oncogenic gene transcription signaling by heterodimerizing with other nuclear receptors viz., LXR or Farnesoid X Receptor (FXR) [10]. This pathway was of interest to us as a potential target due to its role in cell differentiation. Retinoids selectively bind to the RXR ligand binding domain (LBD) and enhance its transcriptional function through RXR homo and/or heterodimers formed with other nuclear receptors. UAB116 is a rexinoid that binds to the LBD of RXR, resulting in conformational changes that promote RXR–LXR heterodimer formation and enhance its signaling [9]. We employed immunoblotting to confirm increased expression of RXR subunits following treatment of HLM_2 cells with UAB116 to verify target activation (Figure 1b).

### 2.2. Effects of UAB116 on Differential Gene Expression in HLM_2 Cells

We next investigated the effects of UAB116 on metastatic HB cells at the genomic level. We used RNA sequencing to evaluate the differences in gene expression in untreated HLM_2 cells compared to HLM_2 cells treated with UAB116 (25 µM) for 72 h. We utilized the IPA tool with parameters of genes showing significant differential expression (*p* < 0.05 with a fold change greater or less than 2) to detect associated biological functions and key canonical pathways affected by UAB116 treatment (Figure 1c). IPA analysis identified activation of retinoids in the UAB116 treated HLM_2 cells (Figure 1c, arrow), corroborating target engagement.

We further analyzed the sequencing data using a threshold of adjusted *p*-value below 0.05 and an absolute fold change greater or less than two and found that UAB116 treatment significantly altered 43 genes (Figure 1d), including 6 differentially upregulated and 37 differentially downregulated genes (Figure 1e). *TRIM29*, which codes for tripartite motif 29, ataxia-telangiectasia group D-associated protein (ATDC), was increased eight-fold in the HLM_2 cells treated with UAB116 as compared to the untreated cells (Figure 1d).

### 2.3. UAB116 Upregulates TRIM29 in HLM_2 Cells

We performed qPCR and immunoblotting to validate the sequencing results. We observed an increase in *TRIM29* mRNA abundance with UAB116 treatment (Figure 2a). The protein expression of TRIM29 also increased following UAB116 treatment (Figure 2b). TRIM29 destabilizes disheveled 2 (Dvl2), resulting in glycogen synthase kinase 3β (GSK3β) activation and a decrease in unphosphorylated cytoplasmic and nuclear β-catenin (both active forms), subsequently inhibiting downstream β-catenin/TCF-regulated targets like c-Myc, cyclin D1, and matrix metalloproteinase-9 (MMP-9) [11]. We performed immunoblotting to evaluate these potential targets in the Wnt/β catenin pathway. Treatment with UAB116 decreased the protein expression of β-catenin in the whole cell extract (Figure 2c). It also decreased the nuclear fraction of β-catenin and the active (non-phosphorylated) form of β-catenin in the cytoplasm (Ser33/37/Thr42) (Figure 2d). The protein expression of c-Myc, cyclin D1, and MMP-9, downstream targets of β-catenin, was decreased following treatment (Figure 2e,f). These data indicate that UAB116 upregulates *TRIM29* mRNA and protein and decreases the expression of β-catenin and its downstream targets.

### 2.4. UAB116 Decreases HLM_2 Proliferation and Affects the Cell Cycle

Since some of the main functions of retinoids are to control proliferation and the cell cycle [12], we evaluated the effects of UAB116 on these entities. We first wanted to determine the cytotoxicity of UAB116 to establish compound dosing for subsequent experiments. A significant reduction in HLM_2 viability was observed 72 h after treatment with UAB116, beginning at a concentration of 50 µM as detected via colorimetric assay with a calculated lethal dose 50% (LD_50_) of 58.4 μM (Figure 3a). Cell counting demonstrated a significant decrease in the number of HLM_2 cells in the group treated with UAB116 (30 μM) at 48 and 72 h compared to untreated cells (4340 ± 460 vs. 1900 ± 200 cells, untreated vs. UAB116, 72 h, *p* ≤ 0.001) (Figure 3b), indicating decreased proliferation. To explore potential mechanisms for the changes in proliferation, we examined the cell cycle. UAB116 treatment led to a failure of progression from G1 to S phase as evidenced by a reduction in the percentage of cells in the S phase following treatment (51.0 ± 0.55 vs. 46.64 ± 0.41, control vs. UAB116, *p* ≤ 0.01) (Figure 3c). There was also an increase in the number of cells in the G2 phase (Figure 3c). Both of these findings are associated with lack of progression through the cell cycle. Immunoblotting indicated that the disruption in cell cycle progression was potentially mediated by the cascade involving the downregulation of CDK 4 and 6, cyclin D1 (Figure 3d), and dephosphorylation of retinoblastoma (Rb) protein (Figure 3e).

### 2.5. UAB116 Decreases HLM_2 Invasion

The β-catenin pathway is responsible for HB cell motility [13]. Therefore, we wanted to evaluate the effects of UAB116 on the invasion potential of HLM_2 cells and utilized modified Boyden chamber assays to complete these studies. Treatment with UAB116 significantly inhibited HLM_2 cell invasion (Figure 4).

### 2.6. UAB116 Decreases HLM_2 Stemness

Previous studies demonstrated that Wnt/β-catenin and Myc signaling leads to an aggressive stem-cell-like phenotype in HB [14]. In an earlier study, we observed that HLM_2 cells were significantly more stemlike than the parent cell line, HuH6 [4]. Our findings, and those of other researchers, prompted us to investigate the impact of rexinoid agonism on stemness. Since HLM_2 cells do not grow as tumor spheres, we performed colony forming assays to assess their stem-like phenotype. UAB116 significantly decreased the clonogenic growth of HLM_2 cells (Figure 5a,b). We examined the expression of common markers of HB stemness with immunoblotting and flow cytometry. Treatment with increasing concentrations of UAB116 (0–50 µM) led to a decrease in the protein expression of Nanog, Nestin (Figure 5c, upper panel), Oct4, and Sox2 (Figure 5c, lower panel). CD133 is another marker of HB stemness [15]. Immunoblotting demonstrated decreased total CD133 protein expression in HLM_2 cells after treatment with UAB116. We used flow cytometry to assess the cell surface expression of CD133 and found that it was significantly decreased after UAB116 treatment (Figure 5e).

### 2.7. Transcriptome Analysis of Published HB Patient Cohorts

We conducted gene expression and correlation analyses on HB patient samples using the R2: Genomics Analysis and Visualization Platform (http://r2platform.com (accessed on 4 April 2024)). Normalized, clinically annotated microarray data were sourced from the GEO dataset GSE131329 [16], inclusive of 67 HB samples, and compared for *TRIM29*, *CTNNB1*, and *RXR* gene abundance. We found an inverse correlation between *TRIM29* and *CTNNB1* gene expression (r-value = −0.613, *p*-value = 3.50 × 10^−8^ ) (Figure 6a). In addition, we observed a positive association between *TRIM29* gene expression and the targets of UAB116 viz., *RXRα* (r-value = 0.466, *p*-value = 6.96 × 10^−5^) (Figure 6b) and *RXRβ* (r-value = 0.308, *p*-value = 0.011) (Figure 6c). These data provide support for the clinical relevance of our findings.

### 2.8. Proposed Mechanism of UAB116 Effect on HLM_2 Cells

Figure 7 depicts the proposed mechanism by which UAB116 may exert its effects on HB. We propose that UAB116 increases LXR/RXR which subsequently acts through upregulation of TRIM29 to turn off Wnt/β-catenin signaling. This decrease in Wnt/β-catenin signaling results in decreased gene transcription of downstream targets including c-Myc, cyclin D1, and MMP-9. The downregulation of these downstream targets could result in the phenotypic changes observed with UAB116 treatment, including decreased proliferation, invasion, and stemness.

## 3. Discussion

This study explored the effects of a novel rexinoid agonist, UAB116, on the metastatic HB phenotype using the metastatic HB cell line, HLM_2. We chose to investigate the LXR/RXR pathway as we found that it was downregulated in these metastatic cells. Similar findings have been reported in other cancers. LXRα expression is decreased in liver cancer, breast cancer, colorectal cancer, lung cancer, myeloma, and sarcomas [17]. Further, LXR activation has been shown to inhibit invasion and proliferation, and to induce apoptosis and cell cycle arrest, normally acting to decrease tumorigenesis [18]. Given these findings, the downregulation of the LXR/RXR pathway in HLM_2 cells provided a rational target for investigation.

LXR/RXR signaling modulates transcriptional targets regulating cell cycle, proliferation, and motility [19] including the Wnt/β-catenin pathway [20,21]. Wnt-mediated β-catenin signaling is involved in the regulation of embryonic development [22] and aberrant activation results in HB tumorigenesis [13]. Several previous preclinical and clinical studies in HB have shown that the activation of Wnt/β-catenin signaling is driven by somatic mutations in exon 3 of the *CTNNB1* (β-catenin) gene, which increases nuclear translocation of β-catenin. The nuclear accumulation of β-catenin induces the transcription of downstream targets, resulting in an increased HB metastatic potential [13]. In our study, we found that activation of the LXR/RXR pathway with UAB116 decreased β-catenin localization in the nucleus and decreased the unphosphorylated fraction in the cytoplasm, both active forms of the protein, and resulted in the down regulation of multiple β-catenin downstream targets, including cyclin D1, c-myc, and MMP-9.

In the current study, we found that treatment with a LXR/RXR agonist upregulates *TRIM29*. Literature reports suggest that the function of *TRIM29* in cancer may be cell line dependent, in that it has been shown to act as both a tumor suppressor or an oncogene [11,23]. For example, in a study of hepatocellular carcinoma (HCC), *TRIM29* acts as a tumor suppressor, decreasing HCC cell proliferation and clonicity [23]. Similarly, in breast and prostate cancer, *TRIM29* overexpression was found to have an inhibitory effect on tumor progression, and those patients with overexpression realized a better prognosis [24,25]. Conversely, in pancreatic ductal cells, *TRIM29* is overexpressed and is linked to an overall poor prognosis [26]. In HCC, the mechanism of *TRIM29* tumor suppression is related to Wnt/β-catenin signaling. Xu and colleagues showed that silencing of *TRIM29* leads to the activation of Wnt/β-catenin signaling, suggesting that *TRIM29* is inhibiting Wnt/β-catenin signaling, thereby functioning as a tumor suppressor [23]. We had similar findings where we noted a decrease in *TRIM29* in metastatic HB cells. Treatment with UAB116 appeared to restore *TRIM29*, as evidenced in genomic studies and by immunoblotting. Identifying a positive correlation between *TRIM29* gene expression and the targets of UAB116, such as *RXRα* and *RXRβ*, in human HB patient samples provided further support for *TRIM29* as a target of UAB116. When *TRIM29* was restored with UAB116 treatment, the active forms of β-catenin were decreased, implying a tumor suppressor function for *TRIM29* that is associated with β-catenin signaling in HB.

Retinoids have been employed as cancer therapies primarily because of their ability to promote differentiation rather than exerting a cytotoxic effect, with the presumption that if the remaining cancer cells could be induced to differentiate into a more benign entity, cancer recurrence may be avoided. In the current study, we found that UAB116 did not affect viability until high concentrations were administered. However, changes in proliferation and cell cycle, and effects on cell cycle proteins, were seen at much lower concentrations. Cancer proliferation is characterized by dysregulation of the cell cycle [27,28]. In the present study, UAB116 treatment was associated with failure of progression out of G2 and from G1/S phase and downregulation of CDK4, CDK6, cyclin D1, and phosphorylated Rb protein, which promote progression through the cell cycle [28]. These findings are similar to those of a previous preclinical investigation of another retinoid agonist, UAB30, in neuroblastoma that showed similar effects on the cell cycle [29]. These findings support the notion that a lack of cell cycle progression contributes to the effects of UAB116 on proliferation.

UAB116 treatment of HLM_2 cells decreased their invasive potential, possibly due to decreased Wnt/β-catenin signaling. β-catenin stimulates the transcription of target genes that promote migration and invasion [30] and dysregulated Wnt/β-catenin signaling contributes to the metastatic phenotype given its role in regulating cell motility, adhesion, and invasion. The decreased protein expression of MMP-9 seen following UAB116 treatment may be another mechanism explaining the decreased invasion noted in this study, since MMPs are induced by Wnt signaling and are required for cell migration [31], invasion [32], and metastasis [33].

Cancer stem cells (CSCs) are a distinct subset of tumor cells characterized by their capacity to exhibit properties similar to normal stem cells [34]. CSCs have been directly implicated in tumor metastasis due to their known properties of pluripotency, dormancy, cellular plasticity, and ability to differentiate, and it has been hypothesized that targeting CSCs may affect the metastatic potential of tumors [35]. Retinoic acid and RXR activation are known to be strong inducers of differentiation in neuroblastoma, glioma, and prostate CSCs [36,37,38] and several preclinical and therapeutic trials have employed retinoids to target CSCs for cancer therapy [39]. In the current study, we used IPA analysis and identified an upregulation of pathways involved in stemness in the HLM_2 cells. We showed that treatment with a retinoid agonist decreased known HB stemness markers [40] including the protein expression of Nestin, Oct4, Sox2, and Nanog, and the cell surface expression of CD133. We were not able to validate stemness with tumor sphere formation experiments as the HLM_2 cells do not grow as spheres, but clonogenic assays provided support for the conclusion that stemness was decreased by treatment with the retinoid agonist.

A notable limitation of the use of retinoid agonists is the known hepatotoxicity and hyperlipidemia that are associated with their use. At this time, bexarotene is the only FDA approved retinoid and its use is limited due to associated hypertriglyceridemia [41]. Novel compounds, such as UAB116, have been designed to limit the off-target effects of hyperlipidemia. Animal studies indicate that hyperlipidemia is reduced by 20% in UAB116 versus bexarotene treated animals [9]. Another limitation of this study is the lack of direct functional validation confirming that *TRIM29* is the primary mediator of the effects of UAB116 on HLM_2 cells via its effect on the Wnt/β-catenin signaling pathway. We provide correlative evidence in the form of transcriptomic analyses and decreased protein expression of β-catenin and its downstream targets following treatment with UAB116. Future studies will incorporate pathway-specific inhibitors or shRNA-mediated knockdown to further establish causality. Additionally, our data show upregulation of *TRIM29* coinciding with decreased β-catenin signaling but we have not yet examined how *TRIM29* interacts with β-catenin on the genomic, transcriptomic, or post-translational levels. *TRIM29* is a member of the E3 ubiquitin ligase family, and future investigations will explore this possible interaction using co-immunoprecipitation and ubiquitination assays [42]. In this study, we did not investigate cell death in a comprehensive fashion, as the primary function of retinoids is differentiation and not cytotoxicity. We believe that UAB116’s suppression of cell proliferation is primarily due to its effect on the cell cycle. Future investigations will address whether apoptosis or other forms of cell death contribute to the changes associated with UAB116.

The current study demonstrated that treatment of a metastatic human HB cell line with UAB116, a LXR/RXR agonist, decreased cell proliferation, invasion, and stemness. We propose that these findings are due to an upregulation of *TRIM29*, subsequently acting to suppress activation of the Wnt/β-catenin signaling pathway. These findings suggest that the use of LXR/RXR agonism warrants further exploration as a potential therapy for patients with metastatic HB.

## 4. Materials and Methods

### 4.1. Cells and Cell Culture

We used the metastatic human HB cell line, HLM_2, which has been previously characterized [4]. The cells were maintained in culture at 37 °C and 5% CO_2_ in Dulbecco’s modified Eagle’s medium (DMEM) supplemented with 10% fetal bovine serum (FBS, HyClone, GE Healthcare Life Sciences, Logan, UT, USA), 1 µg/mL penicillin/streptomycin (Gibco, Carlsbad, CA, USA), and 2 mmol/L L-glutamine (Thermo Fisher Scientific, Waltham, MA, USA). Cells were tested and deemed free of Mycoplasma infection (Universal Mycoplasma Detection Kit, 30-1012K, American Type Culture Collection, ATCC, Manassas, VA, USA). Cell lines were authenticated within the last 12 months through short tandem repeat analysis conducted by the Genomics Core at the University of Alabama at Birmingham (UAB), Birmingham, AL, USA.

### 4.2. Antibodies and Reagents

We employed the following primary antibodies: rabbit monoclonal antibodies against Nanog (3580S), Nestin (73349S), Sox2 (3579S), CDK4 (12790S), CDK6 (13331S), CyclinD1 (2978S), cMyc (18583S), MMP-9 (13667S), Rb (9313S), pRb Ser780 (9307S), pRb Ser795 (9301S), β-catenin (8480S), active β-catenin (8814S), phospho β-catenin (9561S), RXRα (5388S), RXRγ (5629S), TRIM29 (50292S), and anti-vinculin (13901S) from Cell Signaling Technology (Beverly, MA, USA); and rabbit polyclonal anti-Oct4 (19857) from Abcam (Cambridge, MA, USA). The anti-CD133 (18470-1-AP) rabbit polyclonal antibody was obtained from Proteintech (Rosemont, IL, USA), and the mouse monoclonal anti-glyceraldehyde 3-phosphate dehydrogenase (GAPDH, MAB374) and anti-β-actin (A1978) were from Sigma–Aldrich/Millipore (St. Louis, MO, USA).

### 4.3. Viability

Cell viability was assessed using an alamarBlue Cell Viability Assay (Thermo Fisher Scientific). HLM_2 cells (1.5 × 10^4^ per well) were seeded in 96-well plates and exposed to increasing concentrations of UAB116 (0–100 µM). After 72 h of treatment, 10 µL of alamarBlue reagent was applied to each well and the absorbance was measured at an excitation wavelength of 562 nm and emission wavelength of 595 nm using a microplate reader (BioTek Gen5, BioTek, Winooski, VT, USA). The median lethal dose (LD_50_) of the drug was determined.

### 4.4. Proliferation

To evaluate proliferation, HLM_2 (5 × 10^3^ cells per well) were seeded in 12-well plates, allowed to adhere for 4 h, and treated with 30 µM UAB116. After 24, 48, or 72 h, cells were detached using trypsin, stained with 0.4% trypan blue (Gibco), and live cells were manually counted with a hemocytometer.

### 4.5. Cell Cycle

HLM_2 cells (1 × 10^6^) were plated in 4% FBS (low serum) medium, allowed to attach, treated with UAB116 (0–30 µM), and incubated for 24 h. Cells were trypsinized, washed with phosphate-buffered saline (PBS), fixed in cold 100% ethanol, and stained with a mixture of 20 µg/mL propidium iodide (Invitrogen, Thermo Fisher, Eugene, OR, USA) and 0.2 mg/mL RNAse A (Invitrogen) in 0.1% Triton X-100 (Active Motif, Carlsbad, CA, USA). We used an Attune NxT Flow Cytometer (Invitrogen) to obtain data on the cell cycle and analyzed the data with FlowJo software (FlowJo v.10.0.6, LLC, Ashland, OR, USA).

### 4.6. Colony Forming Assays

Colony forming assays were performed by plating 5 × 10^3^ HLM_2 cells per well in 6-well plates. Cells were treated with increasing concentrations of UAB116 (0–50 µM). After 14 days, the cells were fixed with 0.5% crystal violet in 20% methanol for 20 min and later washed with deionized water. Colonies were visualized and counted using ImageJ (National Institutes of Health, NIH, Bethesda, MD, USA) and the Laboratory for Optical and Computational Instrumentation (Madison, WI, USA) (https://imagej.net/ij, accessed on 1 June 2023).

### 4.7. Immunoblotting

Radio-immunoprecipitation assay (RIPA) buffer, mammalian target of rapamycin (mTOR) buffer, nonidet P-40 (NP40) buffer, or nuclear extraction buffers supplemented with protease inhibitors (Sigma–Aldrich), phosphatase inhibitors (Sigma–Aldrich), and phenyl-methane sulfonyl fluoride (Sigma–Aldrich) were used to lyse cells. Immunoblotting, gel transfer, and immunodetection were performed as previously described [15]. The expected molecular weights of the target proteins were verified using the Precision Plus Protein Kaleidoscope marker (Bio-Rad, Hercules, CA, USA). To confirm equal protein loading, we used antibodies against β-actin, GAPDH, or vinculin. We performed densitometry of immunoblots using ImageJ software (National Institutes of Health, NIH, Bethesda, MD, USA) and the Laboratory for Optical and Computational Instrumentation (Madison, WI, USA) (https://imagej.net/ij, accessed on 15 November 2024).

### 4.8. RNA Extraction, Library Preparation, and Sequencing

Total cellular RNA was isolated using the RNeasy kit (Qiagen Inc., Germantown, MD, USA) utilizing the manufacturer’s instructions. Quality control, library preparation, and sequencing were carried out by the UAB genomics core. RNA integrity was evaluated using the Agilent 2100 Bioanalyzer (Agilent Technologies Inc., Waldbronn, Germany) followed by two rounds of Poly A + selection and cDNA synthesis. Libraries were prepared using the NEBNext Ultra Directional RNA Library Prep Kit for Illumina (New England Biolabs, Ipswich, MA, USA) following the provided protocol. Quantification of the libraries was conducted using qPCR in a Roche LightCycler 480 with the Kapa Biosystems kit for library quantitation (Kapa Biosystems, Woburn, MA, USA). Sequencing was completed using the Illumina NextSeq500 platform incorporating the latest versions of the sequencing reagents and flow cells with single-end 75 bp reads.

### 4.9. RNA Sequencing Analysis

The raw RNA-Seq fastq reads were aligned to the human reference genome (GRCh38 p13 Release 43) from Gencode using STAR (version 2.7.11a). Cufflinks (version 2.2.1) was used to estimate transcript abundance using default parameters. Cuffdiff was used to find significant changes in transcript expression, splicing, and promoter usage using the default parameters. For generating network analysis of biological systems, a data set containing gene identifiers and corresponding expression values was uploaded into Ingenuity Pathway Analysis (IPA, https://digitalinsights.qiagen.com/products-overview/discovery-insights-portfolio/analysis-and-visualization/qiagen-ipa/ (accessed on 15 May 2023)) [43]. Each identifier was mapped to its corresponding object using Ingenuity’s knowledge base. To identify significantly differentially regulated molecules, a fold change cut-off of ± 2 was applied. A global molecular network was developed from information available in Ingenuity’s knowledge base and the significantly differentially regulated molecules, called the Network eligible molecules, were overlaid onto this network. Based on the connectivity of these Network eligible molecules, their networks were algorithmically generated. Functional analysis was performed to identify the most significant biological functions of our dataset. To calculate a *p*-value, a right-tailed Fisher’s exact test was used to determine the probability that each biological function assigned to that data set was due to chance alone. We deposited the raw and processed data in Gene Expression Omnibus (GEO), Accession #GSE176152 for HuH6 WT and HLM_2 cells and #GSE290638 for HLM_2 with and without UAB116.

### 4.10. Quantitative Real-Time PCR

Complementary DNA (cDNA) was synthesized using the iScript cDNA Synthesis kit (Bio-Rad) in a 20 µL reaction containing 1 µg of total RNA. Quantitative real-time PCR (qPCR) was performed using SsoAdvanced SYBR Green Supermix (Bio-Rad) according to the manufacturer’s instructions. *TRIM29* primer set 1 forward 5′-ATGCTTGGTGGTCACTTTGG-3′, reverse 5′-GCACTTCCCTTACCAGCATAG-3′ and primer set 3 forward 5′-CTGTTCGCGGGCAATGAGT-3′ and reverse 5′-TGCCTTCCATAGAGTCCATGC-3′ were utilized. Primer specificity was validated using a basic local alignment search as previously described [44]. Each qPCR reaction included 10 ng cDNA in a 20 µL reaction volume. Amplification was performed on an Applied Biosystems 7900HT cycler (Applied Biosystems, Waltham, MA, USA) using the following conditions: 95 °C for 30 s, and then 40 cycles of amplification at 95 °C for 5 s and 60 °C for 10 s. β-actin served as an internal control. Gene expression was calculated using the ∆∆Ct method [45] and is reported as mean fold change ± SEM.

### 4.11. Invasion

Invasion assays were conducted using a modified Boyden chamber technique. The bottom of Transwell inserts (8 µm pores) (Corning Life Sciences, Corning, NY, USA) were coated with collagen Type I (10 µg/mL, MP Biomedicals, Santa Ana, CA, USA) for 4 h at 37 °C, then washed with PBS. The upper surface of the inserts was coated with 50 µL of Matrigel (1 mg/mL, BD Biosciences, San Jose, CA, USA) and incubated at 37 °C for 4 h. HLM_2 cells were seeded in 6-well plates, allowed to adhere, and then treated with UAB116 at concentrations of 0, 25, or 30 µM for 72 h. After treatment, 3 × 10^4^ cells were transferred onto each insert, with conditioned media (350 µL) in the bottom well. After 24 h, the inserts were fixed with 3% paraformaldehyde and stained using 1% crystal violet. Insert images were captured with a light microscope and cell counts were performed in seven random fields per insert using ImageJ software (National Institutes of Health, NIH) and the Laboratory for Optical and Computational Instrumentation (https://imagej.net/ij, accessed on 1 June 2023).

### 4.12. CD133 Expression

We evaluated CD133 cell surface expression using flow cytometry (Invitrogen, Thermo Fisher). HLM_2 cells (4 × 10^5^) were stained with allophycocyanin (APC)-conjugated mouse immunoglobulin G1 (IgG1) anti-human CD133/1 antibody (clone AC133, Miltenyi Biotec, Waltham, MA, USA). Positively stained cells were detected via flow cytometry using the Attune NxT Flow Cytometer (Invitrogen, Thermo Fisher).

### 4.13. UAB116

UAB116, a novel retinoid agonist directed toward LXR/RXR, was designed and produced by our colleagues as described (compound 5 is UAB116) [9]. UAB116 was diluted with dimethyl sulfoxide (DMSO). An equivalent concentration of DMSO, at the highest diluent concentration, was utilized as a control for all experiments and is referred to as control.

### 4.14. Data Analysis

Each experiment was conducted a minimum of three times using independent biological samples. Results are presented as the mean ± standard error of the mean (SEM). Statistical analysis was performed using a two-tailed Student’s *t*-test with *p* ≤ 0.05 considered statistically significant.

## 5. Patents

The University of Alabama at Birmingham has a patent on UAB116, on which VRA is an inventor.

## Figures and Tables

**Figure 1 ijms-26-03933-f001:**
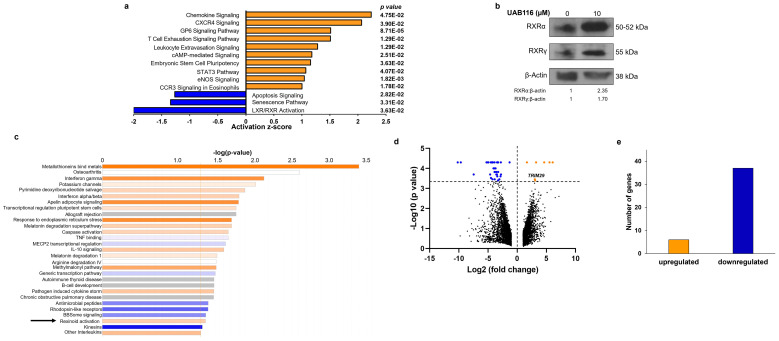
LXR/RXR pathway is downregulated in HLM_2 metastatic human hepatoblastoma (HB) cells. (**a**) Parameters of *p* < 0.05 with a fold change threshold greater or less than two were selected to evaluate the RNA sequencing data of the HLM_2 metastatic HB cells compared to the parent HuH6 HB cells. Ingenuity Pathway Analysis (IPA) of the RNA sequencing data identified canonical pathways that are activated (positive z-scores, orange bars) or inactivated (negative z-scores, blue bars) with the corresponding *p*-value shown to the right of each pathway. HLM_2 cells had a significant upregulation in pathways involved in chemokine signaling and embryonic stem cell pluripotency. In contrast, pathways including LXR/RXR activation, apoptosis, and senescence were significantly downregulated in HLM_2 compared to HuH6 cells. (**b**) HLM_2 cells were treated with UAB116 (0, 10 µM) for 72 h. Protein from whole cell lysates were collected and separated using SDS–PAGE gels. Treatment with UAB116 led to increased expression of RXRα and RXRγ. β-actin served as an internal loading control. (**c**) RNA-Seq results in HLM_2 cells with and without UAB116 treatment were analyzed with IPA, and the top canonical pathways are presented. Activated pathways are represented by orange bars, and inactivated pathways by blue bars. The retinoid pathway was activated with UAB116 treatment (arrow), indicating target engagement. (**d**) Volcano plot of genes expressed in HLM_2 treated with UAB116 compared to untreated cells was constructed from RNA-Seq data. The x-axis displays Log2 fold changes, while the y-axis depicts −Log10 *p*-values. Downregulated genes are shown in blue (fold change < −2 and *p* ≤ 0.05), upregulated genes (fold change > +2 and *p* ≤ 0.05) in orange, and non-significant (NS) genes (with *p* > 0.05) in black. (**e**) Analysis of RNA sequencing data revealed changes in the expression of 43 genes in HLM_2 + UAB116 (25 µM) compared to untreated HLM_2 cells. Sequencing data represent two biologic replicates.

**Figure 2 ijms-26-03933-f002:**
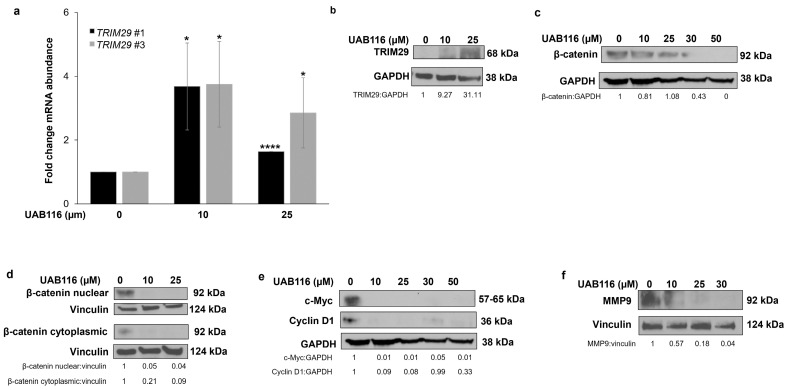
UAB116 upregulates *TRIM29* mRNA and protein in HLM_2 cells. (**a**) HLM_2 (4 × 10^5^) cells were exposed to increasing concentrations of UAB116 (0, 10, or 25 µM) over 72 h. Treatment with UAB116 significantly increased the mRNA abundance of *TRIM29*. *TRIM29* #1 and *TRIM29* #3 represent two different primer sets. (**b**) Immunoblotting of whole cell lysates detected an increase in TRIM29 protein expression after treatment with UAB116. (**c**) Treatment with UAB116 led to decreased protein expression of β-catenin in HLM_2 whole cell lysates. (**d**) Fractionated immunoblotting revealed that nuclear β-catenin and cytoplasmic (active) β-catenin were decreased after UAB116 treatment. (**e**) Protein expression of c-Myc, cyclin D1, and (**f**) MMP-9 decreased after UAB116 treatment. GAPDH and vinculin served as internal loading controls. PCR data are based on a minimum of three biologic replicates, presented as mean ± standard error of the mean (SEM), and were analyzed using two-tailed Student’s *t*-tests. * *p* ≤ 0.05, **** *p* ≤ 0.0001.

**Figure 3 ijms-26-03933-f003:**
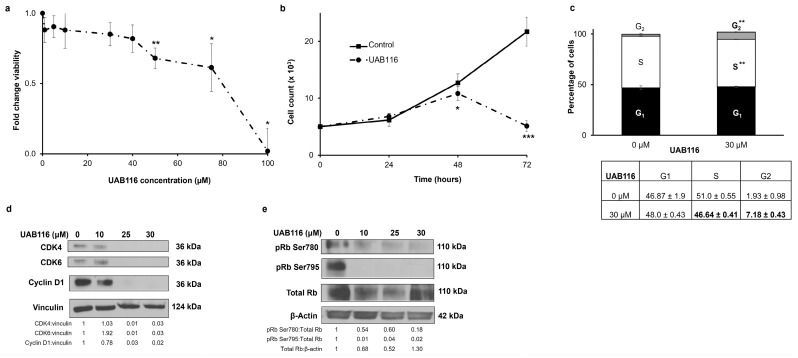
UAB116 decreases HLM_2 proliferation and impedes cell cycle progression. (**a**) Cell viability was assessed with an alamarBlue assay. HLM_2 cells (1.5 × 10^4^ cells per well) were seeded into 96-well plates, given time to adhere, and treated for 72 h with increasing concentrations of UAB116 (0–100 µM). UAB116 reduced cell viability with an estimated lethal dose 50% (LD_50_) of 58.4 μM. (**b**) HLM_2 cells (5 × 10^4^ cells per well) were seeded into 12-well plates, given time to adhere, treated with UAB116 (30 µM), and counted after 24, 48, or 72 h. A significant reduction in cell count was observed at 48 and 72 h after treatment with UAB116. (**c**) HLM_2 cells (1 × 10^6^) were seeded in medium containing 4% FBS, allowed to adhere, and incubated for 24 h with UAB116 (30 µM). Flow cytometry assessed the effects of UAB116 on the cell cycle. UAB116 treatment impaired the transition from G1 to S as indicated by the reduction in the percentage of cells in the S phase (51 ± 0.55 vs. 46.64 ± 0.41, control vs. UAB116 30 µM, *p* ≤ 0.01). Cells did not progress out of G2 phase. Data are depicted in graphical (upper panel) and tabular (lower panel) forms. (**d**) Immunoblotting demonstrated a decrease in protein expression of CDK4, CDK6, and cyclin D1 following treatment with UAB116. Vinculin served as a loading control. (**e**) There was a decrease in the phosphorylation of Rb after treatment with UAB116. Data represent at least three biological replicates, are reported as mean ± SEM, and were evaluated with two-tailed Student’s *t* tests. * *p* ≤ 0.05, ** *p* ≤ 0.01, *** *p* ≤ 0.001.

**Figure 4 ijms-26-03933-f004:**
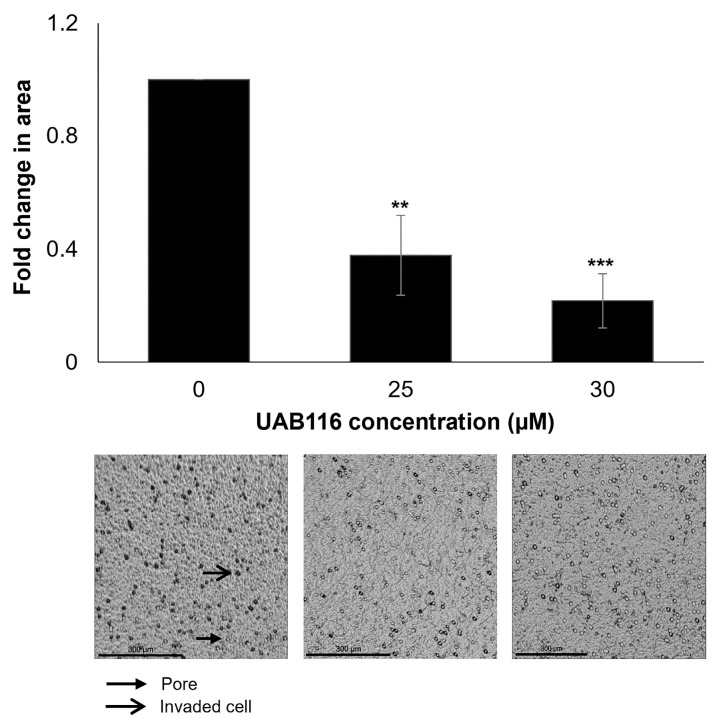
UAB116 decreases HLM_2 invasion. Invasion was measured using modified Boyden chamber assays. HLM_2 cells treated with UAB116 for 72 h were plated (3 × 10^4^) into the top chamber of an 8 µm pore insert coated with Matrigel and allowed to invade for 24 h. Treatment with UAB116 significantly reduced invasion. Representative photomicrographs (10×) of inserts are presented below the graph. Scale bars represent 300 µm. Closed arrow indicates a membrane pore, and open arrow identifies an invading cell. The data denote a minimum of three biologic replicates, are presented as the mean fold change in open area ± SEM, and were analyzed using two-tailed Student’s *t* tests. ** *p* ≤ 0.01, *** *p* ≤ 0.001.

**Figure 5 ijms-26-03933-f005:**
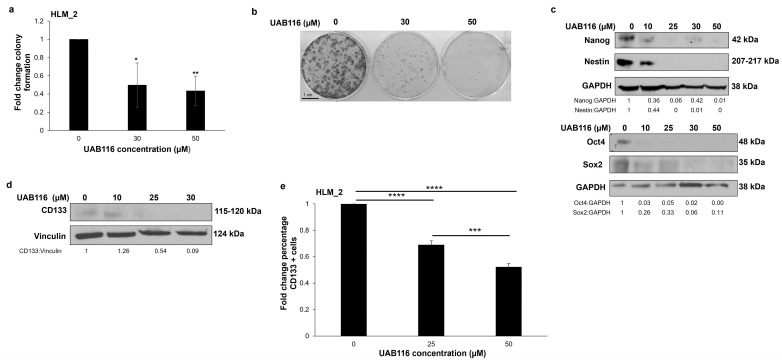
UAB116 affects HLM_2 cell stemness. (**a**) Colony forming assay was used to evaluate the clonogenic potential of HLM_2 cells. UAB116 treatment led to a significant decrease in colony formation by HLM_2 cells. (**b**) Representative photographs of plates from the colony forming assay. (**c**) HLM_2 cells were treated with increasing concentrations of UAB116 (0–50 µM) for 72 h. Immunoblotting of whole cell lysates showing reduced protein expression of stemness markers Nanog and Nestin (upper panel), as well as Oct4 and Sox2 (lower panel). GAPDH served as a loading control. (**d**) Immunoblotting of whole cell lysates showing decreased total CD133 protein expression. Vinculin served as an internal loading control. (**e**) HLM_2 cells were treated with UAB116 (0, 25, 50 µM) for 72 h, stained for CD133, and evaluated with flow cytometry. Cell surface expression of CD133 was significantly reduced following treatment with UAB116. Results are based on a minimum of three biologic replicates, presented as mean ± SEM, and analyzed using two-tailed Student’s *t* tests. Scale bar represents 1 cm. * *p* ≤ 0.05, ** *p* ≤ 0.01, *** *p* ≤ 0.001, **** *p* ≤ 0.0001.

**Figure 6 ijms-26-03933-f006:**
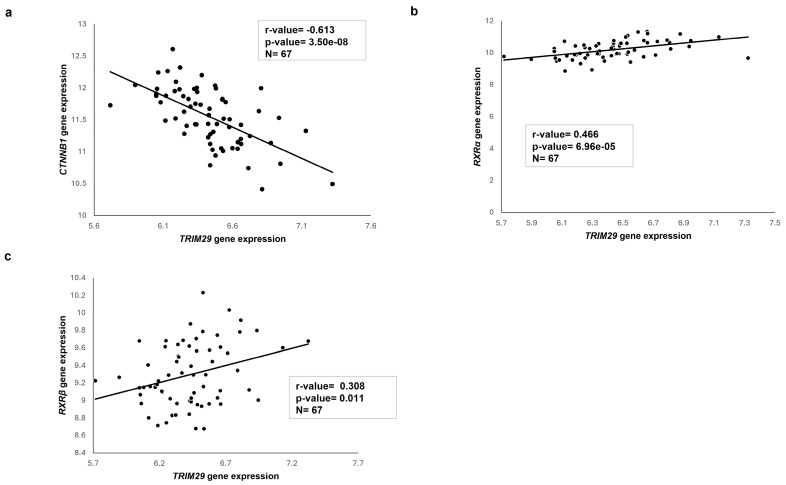
Correlations using the R2 database. The R2 database was queried to evaluate the correlations between *TRIM29* gene abundance and that of several other genes in human HB samples (n = 67). (**a**) There was an inverse correlation between *TRIM29* and the gene most commonly mutated in HB, *CTNNB1* (r-value = −0.613, *p*-value = 3.50 × 10^−8^). (**b**) There was a positive correlation between *TRIM29* and the targets of UAB116 viz., *RXRα* (r-value = 0.466, *p*-value = 6.96 × 10^−5^) and (**c**) *RXRβ* (r-value = 0.308, *p*-value = 0.011).

**Figure 7 ijms-26-03933-f007:**
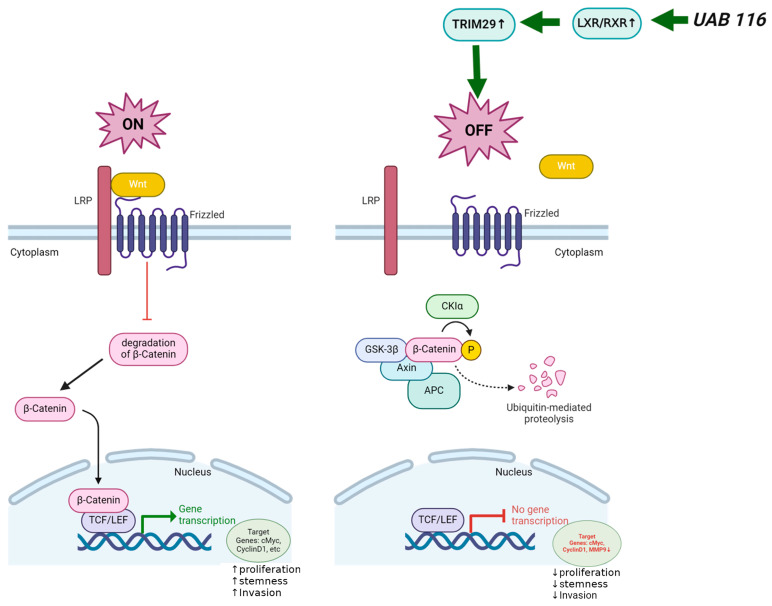
Cartoon depicting the proposed mechanism for effects seen with the rexinoid agonist, UAB116, on HLM_2 HB cells. The left panel depicts the normal status in HLM_2 metastatic HB cells with low *TRIM29* and normal function of the Wnt/β-catenin pathway. The right panel indicates the effect of UAB116, which increases TRIM29 through increased LXR/RXR, thereby deactivating the Wnt/β-catenin pathway, resulting in decreased proliferation, stemness and motility. Created in https://BioRender.com.

## Data Availability

The datasets generated during and/or analyzed during the current study are available in the Gene Expression Omnibus (GEO) (https://www.ncbi.nlm.nih.gov/geo/), Accession #GSE176152 for HuH6 WT and HLM_2 cells and #GSE290638 for HLM_2 with and without UAB116.

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
