# Peer review of "A Novel Rexinoid Agonist, UAB116, Decreases Metastatic Phenotype in Hepatoblastoma by Inhibiting the Wnt/β-Catenin Pathway via Upregulation of TRIM29"

_ijms, 2025, doi:10.3390/ijms26093933_

Round 1

Reviewer 1 Report

Comments and Suggestions for Authors

In the present manuscript, Butey et al. aimed to evaluate the effects of an LXR/RXR agonist, UAB116 on the viability, proliferation, stemness, clonogenicity and motility of hepatoblastoma, and performed RNA sequencing to study differential gene regulation. The author reported that treatment with UAB116 for 72 hours decreased HLM_2 proliferation, stemness, clonogenicity, and invasion. RNA sequencing identified an 8-fold increase in TRIM29 in cells treated with UAB116. At molecular level, the author presented data suggested that UAB116 suppressed HLM_2 proliferation likely through Wnt/β-catenin pathway and the author thus concluded that their data provided support for LXR/RXR agonism as a therapeutic strategy for metastatic HB. Overall the author presented interesting findings that might potentiat clinical applications. This reviewer have suggestions as follow.

  1. The WB results in the current version seems over contrasted, please make corrections throughout the manuscript.
  2. Whether Wnt/β-catenin pathway is the target responsible for UAB116 mediated suppression of HM_2 cell proliferation and invasion need to be confirmed by inhibitor or shRNAs.
  3. In addition, how was β-catenin phoaphorylation and activation, target gene expression need to be addressed in the drug treated cells.
  4. Interactions between TRIM21 and Wnt/β-catenin need to be clarified. Dose TRIM21 interacted with any Wnt/β-catenin components and induced their ubiquitination?
  5. There were clearly decrease of cell counts after 72h treatment, did the author ever check cell death in those cells after treatment?

Comments on the Quality of English Language

The language of the current version is acceptable. 

Reviewer 2 Report

Comments and Suggestions for Authors

The authors have carried out a very interesting study on HB, focusing mainly on its metastatic properties. However, I would like to make a few comments:

  1. The authors mention the relationship between Wnt/β-catenin signaling and hepatoblastoma. In this regard, it is known that WNT regulates PFKFB4 expression (doi.org/10.1242/dev.157644) and that, in turn, PFKFB4 is a metastatic signature in hepatoblastoma (doi.org/10.3390/biom14111394). Have the authors observed alterations in the expression profiles of this gene in their RNAseq? Could you confirm these patterns by qPCR and Wb?
  2. How does PFKFB4 respond upon UAB116 treatment?
  3. In the same trend, have the authors observed LEF1 alterations in HLM_2 vs. HUH6 cells? Does UAB116 treatment have any effect on this gene? LEF1 is an important gene related to the WNT pathway, which is part of the transcriptional fingerprint of several types of childhood cancer (doi.org/10.3390/curroncol32010035).

In general terms, this is a very interesting article, which helps to improve our vision of metastasis in HB, one of the main causes of mortality in this disease. I would like to thank the authors for their efforts and labor. In general terms, this is a very interesting article, which helps to improve our vision of metastasis in HB, one of the main causes of mortality in this disease. I would like to thank the authors for their efforts and labor.

Round 2

Reviewer 1 Report

Comments and Suggestions for Authors

The author addressed all concerns of this reviewer.